# Prediction of the Acute or Late Radiation Toxicity Effects in Radiotherapy Patients Using Ex Vivo Induced Biodosimetric Markers: A Review

**DOI:** 10.3390/jpm10040285

**Published:** 2020-12-16

**Authors:** Volodymyr Vinnikov, Manoor Prakash Hande, Ruth Wilkins, Andrzej Wojcik, Eduardo Zubizarreta, Oleg Belyakov

**Affiliations:** 1S.P. Grigoriev Institute for Medical Radiology and Oncology, National Academy of Medical Science of Ukraine, 61024 Kharkiv, Ukraine; 2Department of Physiology, Yong Loo Lin School of Medicine, National University of Singapore, MD9, 2 Medical Drive, Singapore 117593, Singapore; phsmph@nus.edu.sg; 3Consumer and Clinical Radiation Protection Bureau, Health Canada, 775 Brookfield Road, Ottawa, ON K1A 1C1, Canada; ruth.wilkins@canada.ca; 4Centre for Radiation Protection Research, MBW Department, Stockholm University, Svante Arrhenius väg 20C, Room 515, 10691 Stockholm, Sweden; andrzej.wojcik@su.se; 5Section of Applied Radiation Biology and Radiotherapy, Division of Human Health, Department of Nuclear Sciences and Applications, International Atomic Energy Agency, Vienna International Centre, P.O. Box 100, 1400 Vienna, Austria; E.Zubizarreta@iaea.org

**Keywords:** radiosensitivity, biodosimetry, chromosome aberrations, micronuclei, normal tissue toxicity, radiotherapy, predictive tests

## Abstract

A search for effective methods for the assessment of patients’ individual response to radiation is one of the important tasks of clinical radiobiology. This review summarizes available data on the use of ex vivo cytogenetic markers, typically used for biodosimetry, for the prediction of individual clinical radiosensitivity (normal tissue toxicity, NTT) in cells of cancer patients undergoing therapeutic irradiation. In approximately 50% of the relevant reports, selected for the analysis in peer-reviewed international journals, the average ex vivo induced yield of these biodosimetric markers was higher in patients with severe reactions than in patients with a lower grade of NTT. Also, a significant correlation was sometimes found between the biodosimetric marker yield and the severity of acute or late NTT reactions at an individual level, but this observation was not unequivocally proven. A similar controversy of published results was found regarding the attempts to apply G_2_- and γH2AX foci assays for NTT prediction. A correlation between ex vivo cytogenetic biomarker yields and NTT occurred most frequently when chromosome aberrations (not micronuclei) were measured in lymphocytes (not fibroblasts) irradiated to relatively high doses (4–6 Gy, not 2 Gy) in patients with various grades of late (not early) radiotherapy (RT) morbidity. The limitations of existing approaches are discussed, and recommendations on the improvement of the ex vivo cytogenetic testing for NTT prediction are provided. However, the efficiency of these methods still needs to be validated in properly organized clinical trials involving large and verified patient cohorts.

## 1. Introduction

Radiotherapy (RT) is one of the most effective treatments for cancer. However, it is technically impossible to concentrate the impact of ionizing radiation exclusively on tumors, thus normal tissues, which are included into the treatment plan, are unavoidably exposed [1]. RT is not specific to cancer cells, and radiation-induced cytotoxic effects occur both in the tumor and in normal tissues. The “mode of action” of RT towards tumors and normal tissues, and the respective reasons for the need for well-targeted treatment that delivers the lowest dose possible to the normal tissues and organs, are fully described in special literature [1,2].

In RT there is a wide variation in the reaction of normal tissues, and in many situations, the severity of these reactions limits the dose of RT that can be administered to the tumor [3]. Well documented clinical experience shows that comprehensive normal tissue reactions occur due to differences in individual normal tissue sensitivity. If such variation in normal tissue reactions is due to differences in intrinsic cellular radiosensitivity, it should be possible to predict the former based on the measurement of the latter [4]. Ideally, results of such testing would provide a basis for personalized treatment, i.e., individualizing RT schemes [5,6,7,8,9,10,11,12,13,14,15,16]. Tactics were suggested long ago and include: (i) the identification of rare cases of extreme hyper-radiosensitivity, for which RT should be avoided and replaced by surgery and/or chemotherapy; (ii) the reduction of the total RT dose or the use of alternative fractionation scheme in ‘overreacting’ patients with elevated risk of severe normal tissue complication; and (iii) the escalation of the RT dose for the remaining ‘radioresistant’ patients to enhance the tumor control without an increase in complications [5,6,14]. The goal is strategically valuable because in many cases the individualized RT prescriptions would lead to an increased local tumor control and cure, with unchanged or improved normal tissue complication rates and higher quality life for the patient.

Many studies were carried out in order to establish whether normal cell radiosensitivity correlates with the grade of normal tissue toxicity (NTT) in RT patients [17,18,19,20,21,22,23]. Progress in this research, however, has been hampered by the difficulty in the translation of adverse reactions from the clinic to the laboratory. There is still no unified system for describing normal tissue reactions, and there is still no standard terminology to compare the severity of reactions in different normal tissues. Data comparisons between different radiotherapy centres are complicated by the variety of descriptions of these reactions which are difficult to quantify. It took about 20 years for the most pragmatic terminology proposed in the field of normal tissue radiosensitivity [7] to receive a comprehensive molecular and biochemical explanation [24,25], which only recently has begun to be actively explored in the theory and practice of radiation oncology [20].

In the past 30 years there were many studies that have shown some correlation between ex vivo normal cell radiosensitivity and tissue response to RT, but often these initial conclusions were challenged by the results of larger investigations. In these reports, mainly functional tests such as clonogenic, cell survival or DNA repair assays were applied to determine parameters of cellular radiosensitivity. In addition, several biochemical or molecular end-points have been tested experimentally with the hope of developing systems capable of predicting normal tissue effects to RT. It has become evident that the risk of developing side-effects is predominantly influenced by genetic factors, presumably those regulating DNA repair and relevant pathways [13,14,15,16,26,27,28,29,30,31,32]. Despite huge efforts and long-term research, the real progress in this area has been achieved only recently, when ‘big data’ became available in the framework of large-scale, international projects devoted to the validation of suggested biomarkers of the normal tissue radiotoxicity for clinical use [16,17,20,22,23,25,33,34,35]. Working-out and validation of molecular genetic predictors of radiosensitivity were carried out by consolidated efforts of scientific consortiums in the framework of international programs and projects: EURATOM, Multidisciplinary European Low-Dose Initiative of the European Joint Programme for the Integration of Radiation Protection Research (MELODI/CONCERT), RadGenomics,“Genetic Predictors of Adverse Radiotherapy” (Gene-PARE), “Genetic Pathways for the Prediction of the Effects of Irradiation” (GENEPI), “Assessment of Polymorphisms for Predicting the Effects of Radiotherapy” (RAPPER) or the most recent “Validating Predictive Models and Biomarkers of Radiotherapy Toxicity to Reduce Side-Effects and Improve Quality of Life in Cancer Survivors” (REQUITE) [22,23,26,33,34,35].

Among functional ex vivo assays there is a particular set of biomarkers, which bridges categories of radiation response and genetics. This is radiation-induced cytogenetic (chromosomal) damage, suitable for quantitative analysis and thus applicable to biological dosimetry, i.e., expressing the radiation damage yield in terms of the accumulated radiation dose, considering ‘by default’ the individual radiosensitivity [19]. Unlike molecular predictors based on single nucleotide polymorphisms (SNP) or constitutional gene expression, cytogenetic assays may help to evaluate the radiosensitivity according to the dose and the genetic status.

Some researchers have used cytogenetic methods in this way, but the overall outcomes of these studies appear to be quite controversial. Despite an appreciable number of relevant publications, the analysis of such reports is complicated by variations in the experimental design, end-points andthe ex vivo exposure system used, and also by the heterogeneity of patient groups and normal tissue damage evaluated in these studies. To date, none of the suggested biomarkers has been validated for clinical use as a predictor of the NTT.

The lack of robust approaches to the use of radiation biomarkers for radiation oncology is especially surprising against the background of the growing number of the specialized biodosimetry laboratories in the world in the last decade. That was recognized as a serious problem by the International Atomic Energy Agency (IAEA), and to overcome it the IAEA launched the Coordinated Research Project E35010 MEDBIODOSE, in which the improvement of the methodology of NTT prediction by the measurement of ex vivo cytogenetic damage in patients’ cells comprises an important research task [36,37]. Also, the IAEA organized and coordinated the work of the group of international experts, who performed an extensive analysis of available data on the various aspects of clinical applications of biodosimetry methods. That resulted in the IAEA Human Health Series Report [38], now being prepared for publication. This current article is the supplementary part of the IAEA Human Health Series Report, being focused specifically on the application of biodosimetric markers to test cell radiosensitivity ex vivo in trying to predict NTT in RT patients, along with a discussion of limitations of these approaches. This review is addressed simultaneously to clinical radiation oncologists and radiobiologists focused on biodosimetry to stimulate their interest to collaboration.

## 2. A Brief Overview of Markers Used for the Cytogenetic Biodosimetry

The methodology of radiation biodosimetry has been developed in radiation protection and radiation medicine specifically to deal with scenarios of uncontrolled, accidental overexposure, for which a physical dose reconstruction is unavailable or uncertain. The main principle of the biodosimetry is the estimation of absorbed radiation dose by referring the yield of biomarkers in vivo measured in an exposed person to an appropriate dose response, generated in vitro [39].

The list of necessary characteristics of dosimetric biomarkers includes their specificity to radiation exposure, a clear dose response for a wide range of doses, a possibility to produce a calibration in vitro, and a capability to detect and quantify the heterogeneity of the dose distribution in the human body. Other important traits include responsiveness to the radiation linear energy transfer (LET) factor and an accountable dependence on the protraction or fractionation of exposure and time delay between irradiation and analysis.

Among many biochemical, biophysical, cellular and clinical end-points, tested as potential radiation dosimetric markers, only a few were found to be suitable for this purpose, and all are based on cytogenetic damage observed in metaphase spreads of cultured human cells. These include:dicentrics and centric rings (Dic+CR);stable chromosome exchanges (mainly, translocations (Tn)) and complex chromosomal rearrangements (CCR) visualized using a fluorescence in situ hybridization (FISH) technique;fragments and/or rings quantified among prematurely condensed chromosomes (PCC);micronuclei (MN) scored in cytokinesis blocked binucleated cells.

Dic+CR, PCC rings and fragments and MN are considered as ‘unstable’ biomarkers, because they are eliminated through cell divisions. The essential proportion of Tn are transmissible to daughter cells, but not all of them. Therefore the quantitative measurement of the radiation-induced yield of cytogenetic damage must be restricted to the 1st post-radiation mitosis. That is achieved by identifying the cell cycle number of each metaphase, if Bromodeoxiuridine (BrdU) is added to a cell culture. This technique is not applicable to the MN assay, thus scoring of micronuclei must be carried out in binucleated cells only.

Also, phosphorylated histones γH2AX, representing sites of DNA double strand break repair in interphase lymphocyte nuclei (γH2AX foci), can be used for biodosimetric purposes, but with strong limitations on time scale. The γH2AX foci appear within minutes of exposure and increase until about 1 h, after which they decline back to near background levels. This rapid change limits their ability for accurate dosimetry but still allows their use for an indication of exposure and their applicability for the radiosensitivity assessment testing in well-controlled experimental conditions.

Human peripheral blood lymphocytes (PBL) are now preferred for assays based on radiation biomarkers, particularly, as an alternative to fibroblasts, because of the ease of obtaining samples and the rapid results that can be generated. Technical aspects of the cytogenetic biodosimetry are well refined [39,40] and have reached the level of the international standard (see [37] for review). Advanced cytogenetic biodosimetry methods provide the results formalized as a mean dose of radiation of a certain quality absorbed in a certain volume fraction of lymphocytes, both values supplied with respective confidence intervals.

All existing external RT regimens, including total and partial body irradiation, most therapeutic radionuclides and diagnostic radiology procedures cause a considerable increase of the yield of bioindicators used in biodosimetry in patients’ PBL (summarized data of the literature on this issue will be published as the IAEA Human Health Series Report [38]).

The use of radiation dose response biomarkers for practical purposes in radiation oncology and medical radiology has several clear advantages. These markers are recognized as measures of genotoxicity due to their relationship to DNA-damage response and, therefore, may be used for direct assessment of the risk of radiation-related pathology, including both deterministic and stochastic effects. All validated biomarkers show strong dependence on radiation dose, dose rate, irradiated volume [39] and thus theoretically can respond to the impact of radiation modifiers. Therefore, such dosimetric biomarkers can be considered as an effective tool for quantifying individualized radiation load in patients, either in vivo or ex vivo.

The theory for the application of cytogenetic damage for the prediction of normal tissue response to radiation is based on three main points. First, chromosomal aberrations (ChA) are the direct outcome of DNA repair, which is one of the core determinants of cellular radiosensitivity. Studies on DNA repair-deficient cells have demonstrated that any alterations in DNA repair genes or malfunctions of their proteins can have a significant impact on the ChA yield [41,42,43]. Second, some ChA (e.g., unbalanced exchanges and fragments) are the direct cause of mitotic cell death [44,45] and are, therefore, mechanistically linked to clonogenic cell survival [46,47]. Consequently, it can be expected that ChA (or their precursors—DNA breaks, or their products—MN) would also correlate with the tissue repair maintained by cell proliferation, and hence with the time of on-set and the severity of radiation-induced NTT. Third, in studies on cells of mono- and disigotic twins a significant hereditary impact on the manifestation of individual cellular radiosensitivity had been demonstrated for various end-points, including ChA or MN [29,48,49,50,51,52,53,54,55].

Combining all three reasons, the extrapolation from the cytogenetic damage induction, as a genetically controlled trait, to a radiosensitive phenotype seems to be reasonable.

## 3. Review of Existing Studies

Attempts to apply biodosimetric markers to the prognosis of normal tissue response to RT have been made many times. The main hypothesis tested in this research was that cancer patients with a higher yield of ex vivo radiation-induced cytogenetic damage in lymphocytes or other cells are more likely to develop RT-related morbidity. Three main biodosimetry end-points were used forex vivo testing of the patients’ cells: Dic+CR, MN, and Tn or CCR visualized by FISH. Occasionally PCC methods were also applied for assessing chromosome damage, and in recent years the quantification of DNA double strand breaks (DSBs) by γH2AX foci analysis has been actively introduced into practice. In these studies, typically, cells from patients are irradiated ex vivo to X-rays or γ-rays with different radiation doses and dose rates. Some researchers have used several biodosimetric techniques in one study, e.g., dicentrics, translocations and γH2AX foci, in an attempt to determine the best end-point for predicting clinical radiosensitivity [56,57,58]. Among various cellular test-systems used for the assessment of chromosomal radiosensitivity, human PBL appear to be the most appropriate due the lack of a need for long cell culture periods, standardized culturing conditions and a well-developed methodology of cytogenetic marker quantification.

### 3.1. Cytogenetic Expertise of Individual Cases Showing Elevated Normal Tissue Toxicity (NTT)

In clinical practice of radiation oncology, the unexpected, extreme tissue reactions are still frequently attributed to the malfunction of radiotherapy devices or erroneous dosimetry. In such situations expertise is needed to clarify the reason: the accident or patient’s intrinsic over-reactive status. It should be noted that, from the beginning, cytogenetic analyses were highly informative for confirming the radiosensitivity post factum in individual cases as identified by clinical symptoms. As a rule, patients with abnormal tissue radiosensitivity displayed a higher outcome of ChA per unit dose induced in lymphocytes by ex vivo irradiation [59,60,61,62]. Thus, radiation biomarkers can be used effectively for revealing a possible mechanism of radiosensitivity, particularly a malfunction of the DNA repair machinery involved in the ChA formation. Results from cytogenetic testing have led to an active exploration of tests based on quantifying DNA damage repair functionality, primarily the kinetics of DNA DSB, assessed by specific methods like γH2AX foci, which will be discussed later.

Importantly, elevated rates of ChA induced in vitro have been observed in clinically radiosensitive patients who did not have any constitutional chromosomal abnormalities [59], known genetic disorders with increased radiosensitivity [60,62], or apparent hereditary chromosome instability, cancer prone syndromes [61]. The relationship between the two latter categories is not simple. Genetic syndromes, which are linked to radiosensitivity, are generally associated with high cancer risk, while patients with syndromes linked to increased rates of tumor occurrence, are not necessarily have radiosensitive cells or tissues [18]. Nevertheless, many authors consider a high number of ex vivo induced ChA or MN as a hallmark of a genetically based radio-sensitive phenotype in patients showing enhanced tissue radiosensitivity.

### 3.2. Group Studies of Ex Vivo Cytogenetic Response in Cells of Patients with Various NTT Grades

In contrast to individualized studies, which focused on one or few patients, those cytogenetic predictive assays, which were performed in larger, randomly formed groups of patients and especially in a prospective, screening mode, gave rather ambiguous results.

Table 1 and Table 2 summarize briefly the data obtained by different research groups in their attempts to correlate G_0_ cytogenetic radiosensitivity with normal tissue reactions in RT patients. The non-exhaustive list includes 16 reports based on ChA analysis (Table 1) and 16 studies using MN test (Table 2). It can be seen that the study design varied considerably between laboratories. The most common tumor location studied was breast cancer, with head and neck, prostate and gynecological cancers being the next most popular. About half of the publications (17 out of 32) presented retrospective studies, while the prospective studies were less frequent (11 out 32), and 4 reports contained both types of studies. Furthermore, early/acute RT reactions of normal tissues and organs were considered in 9 studies, late NTT toxicity were the focus of interest in 12 reports, and 11 publications contained data on both categories of clinical effects.

In 6 studies, positive controls (i.e., matched patients with no normal tissue damage) were not specifically included, so that the ex vivo induced cytogenetic damage yield in overreacting patients was compared with that of healthy donor cells [63,64,73], or between overreactors with different NTT types [84], or with the yield in a single patient with no toxic reactions at the time of study [91,92]. Although such results have some academic interest, they cannot be fully used for analysis.

#### 3.2.1. Efficacy of Different End-Points

Considering acute and late NTT effects separately, and excluding study [84] from the score, the distribution of the results in 41 analyzed datasets (22 with on ChA and 19 with MN data) is as follows. A positive correlation between cytogenetic damage induced in PBL ex vivo and clinical radiation toxicity grade was observed in 11 reports with acute reactions (6 using ChA and 5 using MN assay) and 14 reports involving late effects (10 using ChA and 4 using MN assay). In one of these studies, the correlation occurred in a subgroup of patients, initially selected for a high level of induced residual DNA damage measured using a ‘comet’ assay ex vivo. In this case, the MN test was a secondary discriminative tool within a multiparametric approach [85]. A significant correlation between the ex vivo frequencies of Tn and the latency of skin side effects but not with the grade of the effect was observed in another study [74]. Nevertheless, these results demonstrated a rather high sensitivity and specificity. In two reports claiming a positive correlation between the ex vivo MN yield and the severity of adverse NTT effects [91,92] both research groups did not subtract the background MN frequency from the yield observed in irradiated samples prior to correlation analysis, therefore the authors’ conclusions are rendered suspect. Also, in one of the most recent studies the inverse (!) correlation was found between the NTT grades and MN yields induced by ex vivo radiation doses 2.0 and 10.0 Gy; no explanation for this phenomenon was provided [93].

The absence of a link between cytogenetic and clinical radiosensitivity was shown in 15 datasets. That included 6 studies based on ChA analysis: 2 reports contained respective data on acute effects, and 4 on late NTT. Among 9 observations based on MN assay, 6 were focused on acute and 3 on late NTT effects. Interestingly, one of the research groups, which found no correlation between the induced MN yields and acute NTT grades in breast cancer patients, reported quite different MN yields in their two consequent publications [89,90], and the reasons for such a discrepancy are unclear.

Thus, in general, the analysis of ChAs in conventionally stained or FISH-painted metaphases or PCC spreads appeared to be more suitable for clinical ex vivo irradiation tests compared to the MN assay. Among various types of chromosomal damage registered during the analysis, chromosome fragments (excess acentrics or PCC fragments) and color junctions or CCR detectable by FISH were the most sensitive ex vivo markers for acute NTT effects or late lesions.

No clear mechanistic explanation for the predictive ability of chromosome fragment-type breakage has been suggested yet. Meanwhile, the elevated numbers of CCR observed in metaphases of clinical over-reactors may be linked to defects in one or several of the cell-cycle checkpoints, which when functioning normally, prevent heavily damaged cells from entering mitosis [71].

Besides a compromised cell cycle regulation, cytogenetic radiation biomarkers, which are in the focus of current review, have strong mechanistic linkage to the complex interplay of several DNA DSB repair pathways, including their cell cycle-specific impairment, which causes a shift from the canonical non-homologous end-joining (NHEJ) or homologous recombination (HR) towards alternative, more error-prone mechanisms [94,95,96]. In line with this, it was shown that genes of both HR and NHEJ pathways in PBL of severe reacting patients were less induced by ex vivo irradiation in compare with that in patients without late reactions [97]. Obviously, acell cycle-dependent analysis of DSB repair may be valuable for the expression of clinical hypersensitivity to ionizing radiation, e.g., as shown by Zahnreich et al. [98]. However, the overall problem of the correlation between radiosensitivity and DSB repair is very broad, and its discussion is beyond the scope of our review.

#### 3.2.2. The Role of Dose and Dose Rate of Ex Vivo Exposures

It should be noted, that in two studies using the MN test [77,87] and in one using the ChA test [75], which did not detect a predictive value of cytogenetics, the irradiated lymphocytes were cultured longer than is normally done according to biodosimetry standards [39] and without an indicator for cell cycle (such as BrdU for ChAs), even though the doses used for ex vivo irradiation (2–3.5 Gy) were not high enough to cause a significant mitotic delay. The absence of correlation between cytogenetic and clinical radiosensitivity in these reports can, therefore, be partially attributed to aberration-free cells surviving into 2nd divisions while damaged cells would more likely be eliminated in the 1st division. There are other studies in the list, which also involved similar longer-term cell culturing, but in which a correlation was found. In these reports the radiation dose (6 Gy) was high enough to produce both a mitotic delay and aberrant status in 100% of irradiated lymphocytes [67,69]. However, due to the absence of BrdU in cultures it cannot be excluded that some chromosome damage actually occurred in the 2nd mitoses (M2 cells), so perhaps the more damaged cells were being lost and the less damaged cells passed through to M2. This methodological defect may essentially contribute to the overall heterogeneity of the estimates of individual chromosomal radiosensitivity.

One important question is: which radiation doses and dose rates should be used for ex vivo irradiations? In all reports, in which biodosimetric markers showed a notable predictive value, the authors used relatively high ex vivo radiation doses (6 Gy for Dic+CR method and 2–4 Gy for FISH and MN assays). It seems that the approach ‘The Dose must be As High As Possible’ (DAHAP) is applicable here. From the point of classical biodosimetry, such doses are the highest normally used for constructing calibration curves in vitro [39]. After 6 Gy of γ- or X-rays, all metaphase cells carry unstable aberrations, but accurate quantification of chromosome rearrangements is still possible ([99] and references therein). By using caffeine or other chemicals that overcome the G_2_/M block, or by applying a PCC method, higher doses might be attainable. However, this will also artificially increase the yield of chromosome rearrangements being visible in cells, which normally would have been arrested and dead. If the correlation between ex vivo ChA yields and NTT is somehow relevant to the inter-individual differences in the G_2_/M block, then its intentional abrogation may lead to the lesser heterogeneity of this radiobiological index in the patients group, and thus reduce the separation power of the assay, worsening the overall prediction of the risk of NTT.

Also, with FISH painting, the total detectable yield of ChAs per unit dose is much higher than with conventional solid stain analysis due to inclusion of Tn and CCR, thus aberration scoring becomes difficult at doses above 4 Gy. Similarly, the MN yield in binucleated cells reaches a plateau and shows a faulty dose response at doses higher than 5 Gy [100,101]. It should be noted that 10 Gy, used in the study with an unusual result [93], essentially exceeds the upper dose threshold of the practical application of the MN assay in human PBLs. However, the lower doses, particularly <2 Gy, did not provide enough discrimination of an ex vivo effect among patients.

The role of the radiation dose can be illustrated by the data presented by Borgman et al. [67]: the number of chromosomal deletions induced ex vivo was plotted as a function of dose and, although there was already some inter-patient variation at 3 Gy, it became clearer at 6 Gy. Importantly, there was a poor correlation between the aberration yields at the two doses, and as a consequence, the classifications between resistant, normal, and sensitive patients obtained at 3 and 6 Gy were not identical. This discrepancy can be partially explained by the lack of a cell cycle control using BrdU that would cause some unknown proportion of the 2nd division metaphases to be included in the analysis at 3 Gy, contributing to intra-individual heterogeneity, while at 6 Gy this effect would be much lower. The overall problem of reproducibility of ex vivo data will be discussed in more detail below, but regardless of the mechanism, these results showed that the association between individual cytogenetic radiosensitivity and NTT risk can be different for different dose levels. The authors [67] concluded that in order to obtain a robust discrimination between the radiation responses of patients a sufficiently high dose is required. This is a prerequisite for detecting a clear association with the risk of clinical effects.

Previously, a similar conclusion was made in the study [80], in which the authors compared complete dose responses (0–4 Gy) generated for MN yield ex vivo in prostate cancer patients: “We believe that an assessment of individual intrinsic radiosensitivity at only one radiation dose level can be misleading, and that the accurate discrimination of individual radiation sensitivity differences necessitates the determination of the dose–response from baseline (0 Gy) to 4 Gyex vivo irradiation”. This theory was supported by the comparison of the entire area under the curve (AUC) of the dose response generated for MN in the dose range 0.2–3 Gy in a case-control study [88]. Using this parameter, 10 out of 12 NTT cases scored higher than their matched controls, however, 6 of the 12 pairs showed overlap in their standard deviations.

Regarding the dose rate, the majority of studies involved an acute irradiation performed at high dose rates (HDR). Based on the experience of non-cytogenetic, cellular testing, as well as from cytogenetic research unrelated to NTT on cells of healthy individuals, carriers of DNA-repair-deficiency syndromes and cancer patients, it is known that the low dose rate (LDR) approach allows better discrimination (stratifying) of patients according to their intrinsic cellular radiosensitivity [63,102]. Nevertheless, only one NTT-related report was found that used a LDR for the ChA assay [63] and one for the MN assay [77]. In the former study, the LDR approach revealed a difference between over-reacting patients and healthy donors, however, the fact that no normal RT patients were included casts doubts on these conclusions. Furthermore, in the latter study, the LDR approach was not conclusive. It should be kept in mind that the main advantage of using a LDR is the dose rate sparing effect, which can be estimated only in comparison with HDR exposure results. Ideally, both ex vivo dose rates would be used for each patient, doubling the resources needed. The necessity of keeping cells at physiological conditions (temperature of 37 °C and 5% CO_2_ atmosphere) during a prolonged irradiation time also makes the LDR method more technically demanding. These considerations highlight the unsuitability of the LDR approach for NTT predictive testing in clinical practice. The HDR approach provides meaningful results with much higher success, especially when the correct methodology of the NTT data analysis is used.

#### 3.2.3. Clinical, Methodological and Statistical Confounders

There are three parameters to be considered in view of RT induced normal tissue damage: the severity (grade), the frequency and the onset time or latency period for its occurrence. The NTT grade is the basic factor by which patient groups are stratified in retrospective studies, and for which the correlation with ex vivo cytogenetic radiosensitivity is usually evaluated. Only a few reports showed a linkage between ChA or MN yields and the frequency of NTT cases [67], or the actuarial rate of NTT occurrence [66], or the latency period of NTT development [68,74]. It has become apparent that the validity of cytogenetic predictive tests should be defined by stratifying patients according to their chromosomal radiosensitivity, followed by the comparison of the predicted versus observed clinical radiation sequelae. This might also explain why several studies, which stratified according to the observed NTT effect instead of to the cytogenetic test result, failed to demonstrate an association between individual cellular radiosensitivity and NTT. By contrast, in the listed studies that stratified groups according to predicted radiosensitivity, the conclusions about the predictive value of ex vivo tests were the most accurate.

Individual radiosensitivity, assessed by induced cytogenetic damage yield ex vivo, was usually described by normal distribution of individual levels of ChA or MN, exactly as expected from the stochastic nature of chromosomal rearrangements [66,67,75,81,103,104]. However, not all the studies of the possible NTT predictors included statistical analysis of that distribution. Sometimes it was shown that cancer patients, especially those who showed elevated NTT grades, had a much broader spectrum of aberration yields per donor when compared to healthy individuals (e.g., [75]).

Also, in contrast to standard biodosimetry methodology, the ChA or MN per cell distribution was rarely tested for consistency with expected Poisson statistics in ex vivo NTT-related radiosensitivity studies. Moreover, if it was done, any significant over dispersion was not explained [75]. Meanwhile, retrospective studies involve taking blood from patients, who were irradiated in the past and thus carry a certain elevated ‘baseline’ yield of radiation-induced ChA or MN, as compared to normal spontaneous level in healthy control donors. In most such reports the ChA or MN frequencies observed in ex vivo irradiated cells were corrected for the frequencies in unirradiated cells. Usually, the total number of ChA in the control samples was subtracted from the total number of ChA in the irradiated ones without controlling for the specific type of aberration (e.g., in [75]). This ignores the fact that different types of chromosomal rearrangement make different quantitative contributions to spontaneous levels, RT-induced ‘baseline’ yields or ex vivo induced aberrations. Moreover, in particular for FISH-based testing, subtracting the baseline yield brings a lot of uncertainty due to the presence of metaphases with multiple aberrations in RT patients. While it is well known from the biodosimetry practice that the inclusion or rejection of just one or two such cells during the analysis may substantially change the overall aberration yield [105].

The FISH-based end-point, CCR, appears to be especially vulnerable to confounding factors: Lymphocytes of patients having just undergone RT exhibit high baseline frequencies of CCR, which are also dependent on the time since the previous RT and influenced by previous cytostatic therapy. The most important factor, however, is that cells taken from patients during or after RT may respond to ex vivo irradiation with a more drastic increase of CCR than lymphocytes of non-exposed patients [71]. Even the most successful parameter identified to date for predicting NTT, i.e., the proportion of breaks involved in CCRs, which according to [71] are not affected by previous cytostatic treatment and the magnitude of ex vivo dose, should be treated with caution, because their quantification may be affected by the scoring system applied in the study: Protocol for Aberration Identification and Nomenclature Terminology (PAINT) or Savage and Simpson (S&S) nomenclature. Thus, when planning the research and interpretation of results in terms of the linkage between cytogenetic radiosensitivity and clinical NTT, such factors have to be considered carefully.

It should be noted that the general methodology of ex vivo testing is far from complete and cohesive. There have been specific studies addressing the question of which cytogenetic parameters are the most suitable for discriminating patients with increased chromosomal radiosensitivity from healthy individuals [73], and how many metaphases need to be analyzed [106]. However, there has been no such study focused specifically on patients with different grades of clinical radiosensitivity. Furthermore, in NTT studies, the numbers of cells scored at different ex vivo radiation doses were chosen arbitrarily, e.g., 200 metaphase spreads were scored for chromosomal aberrations at 2 Gy, 400 metaphases at 0.7 Gy and 1000 metaphases at 0 Gy [71,72]. These studies did not take into account the recommendations for biodosimetry [39] for either the optimal number of metaphases scored (500 for conventional and 1500 for FISH analysis), or for determining the required accuracy of the estimate (the ratio of the error to the yield) based on the Poisson statistics for the aberration mean yield and per-cell distribution.

The main issues are the intra-individual heterogeneity and overall reproducibility of ex vivo testing results, particularly because the biodosimetric markers measured in these NTT prediction studies are stochastic radiation effects, which show a certain natural variability. Unfortunately, examples of systematic repeated testing of RT patients’ cells, which are needed to examine this natural variability, are rare. In one study, samples from seven patients were analyzed two or three times after RT and showed a stable general pattern of cytogenetic reaction, including CCR induction [71]. In a second study, repeat samples were tested in 13 patients with the time between sampling ranging from 3 to 9 months. Good reproducibility of the HDR MN assay results was demonstrated by a strong correlation between the repeat samples [77].

Among other RT patient-related publications, the reproducibility of the ex vivo assay has been mentioned twice, but both times with respect to blood samples taken from healthy donors [81,103]. Thus, a quality assurance and quality control (QA/QC) system still needs to be developed for the area of clinical use of ex vivo radiation biomarkers for NTT prediction in RT patients, starting with basic validation steps: sensitivity, specificity, reproducibility, confounders.

### 3.3. Studies Using Non-Lymphocyte Cell Systems

In trying to use a cytogenetic test-system closely linked to cell survival assays, some researchers have measured the yield of radiation-induced biomarkers in cultured skin fibroblasts, keratinocytes or lymphoblastoid cell lines.

An extensive analysis of ex vivo MN was performed in cultured skin fibroblasts of 17 patients with increased acute and/or late side effects along with 10 patients with no excessive reactions [107]. Dose response curves were generated individually for each patient in the range of 1–7 Gy, however a saturation or decrease in MN yield at doses ≥4 Gy occurred nearly in all cases. The cells of the majority of the sensitive patients showed a higher MN induction than the average of the donors with a normal response. Only two of the patients with acute reactions and four with late effects had a dose response clearly below or similar to the average of the normal patients.

In a study performed on fibroblasts of 8 retrospectively examined patients with cervical or head andneck cancers, no significant correlation was found between the rate of ex vivo MN induction in fibroblasts (2–5 Gy) and acute and late normal tissue reaction scores [78]. In addition, no relationship was observed between the ex vivo cytogenetic radiosensitivity of lymphocytes and fibroblasts derived from the same individuals in this work (6 cancer patients plus 5 healthy donors).

Dermal fibroblast lines were established from skin biopsies of 26 patients with soft tissue sarcoma and subjected to 2.4 Gy of low dose-rate (0.0194 Gy min^−1^)^60^Co γ-rays [108]. The MN frequency in irradiated fibroblasts did not correspond to differences in normal tissue responses, which were wound-healing complications and subcutaneous fibrosis.

Later a more sophisticated study was performed in order to compare the dose responses for MN in cultured primary fibroblasts (2–4 Gy γ-rays in vitro) and long-term lymphocyte cell lines (1–2 Gy γ-rays in vitro) derived from 36 patients who had severe acute or late reactions from RT [109]. Heterogeneity of MN frequency in irradiated fibroblasts and lymphocyte cell lines (LCLs) was apparent. Across the different doses, the average MN frequency consistently trended towards being higher in cells obtained from clinically radiosensitive individuals versus those of normal responders (controls). Also, in separately examined subgroups of LCLs derived from patients who had breast cancer, the severe acute reactors showed a significant difference of the average number of cells with multiple MN compared with controls. Among 7 paired fibroblast lines and LCLs derived from the same clinically radiosensitive patients, only one individual with late reactions showed a significant correlation between the two cell lineages for their radiosensitivity, presenting as very high MN frequency.

Also, in a perspective study of 32 cervical cancer patients, an ex vivo MN assay dose response (0.05–4 Gy ^60^Co γ-rays) in fibroblasts and keratinocytes was compared to the normal tissue reactions [110]. Despite the presence of 6 patients with a hyper-radiosensitivity (HRS)-like ex vivo response, the radiation-induced MN did not correlate, either in fibroblasts or keratinocytes, with the grade of acute or late reactions in patients. Five of the 6 patients with HRS cells did not suffer from any mild or severe side effects after RT. Thus, the MN assay showed no predictive value.

The most recent and the largest study to date aimed at establishing possible quantitative links between RT-related overreaction grades and MN yield induced ex vivo (2 Gy γ-rays) in fibroblasts. The study involved more than 100 patient skin biopsy specimens [25]. The MN yield remaining 24 h post-irradiation discriminated three patient subpopulations: radioresistant, overreacting and hyper-radiosensitive patients as classified using the Common Terminology Criteria for Adverse Events (CTCAE). These sub-populations corresponded to three groups of DNA-repair based radiosensitivity defined initially in that study which, by surprise, appeared to be in line with the pragmatic clinical classification [7]. However, within the overreacting cohort the MN test could not discriminate between patients with different clinical radiosensitivity, whether classified using the CTCAE or RTOG scales. These results suggest that ex vivo radiation-induced MN can only distinguish large differences in radiosensitivity.

Based on these studies, MN analysis in fibroblasts, keratinocytes and LCLs does not seem to provide a strong predictive value for radiosensitivity. In addition, these assays take a long time to conduct, requiring cells to be grown from biopsies or transformed from lymphocytes. Therefore, these assays are not effective or practical for the clinical setting. As shown above, PBL have higher potential to be a more appropriate test-system for cytogenetic research aimed at the assessment of chromosomal radiosensitivity for NTT prediction.

### 3.4. Prediction of NTT Using Ex Vivo Tests Based on Other DNA or Chromosome Damage Biomarkers

There are three radiation-induced cellular effects which are promising ex vivo irradiation assays for predicting patient clinical radiosensitivity: γ-H2AX foci, which appear in response to DNA DSBs, chromatid aberrations induced in the G_2_ phase of the cell cycle and alterations of the length of telomeres. These radiation biomarkers are not adapted in classic biodosimetry, thus are subjected only to very brief analysis in the current review.

#### 3.4.1. γ-H2AX Foci

A brief overview of the state of the art of using γ-H2AX in clinical settings has been presented by Redon et al. [111]. The induction of γ-H2AX foci is directly related to DNA DSB recognition and repair, thus a possible relation of their initially-induced or residual yield to clinical radiosensitivity should be considered along with other DNA repair-based assays (e.g., Comet assay). The number of publications highlighting the possibilities and limitations of surrogate end points based on DNA damage repair as predictors of NTT far exceeds the limits of this review. Nevertheless, γ-H2AX foci is claimed as a useful tool in triage biodosimetry and recommended for inclusion into the toolbox of radiation cytogenetic laboratories [112,113,114,115,116], thus it is appropriate to present a brief analysis of the predictive value of this particular end-point.

In our non-exhaustive list of publications about ex vivo induced γ-H2AX foci yield in isolated PBL in patients with various NTT effects there are 13 reports showing that quantification of γ-H2AX foci by microscopy or flow cytometry is not predictive of acute or late radiation toxicity [58,87,88,117,118,119,120,121,122,123,124,125,126]. On the other hand, there are a number of reports showing the opposite result. Earlier, there was a report about a patient who had previously shown severe side effects after RT, and whose lymphocytes in vivo displayed levels of γ-H2AX foci at various sampling times after Computed Tomography (CT) that were several times higher than those of normal individuals. Furthermore, fibroblasts from the same patient also showed significant ex vivo radiosensitivity by γ-H2AX foci analysis [127]. More recently, ex vivo testing of lymphocytes by this technique has shown remarkable differences between groups of patients with high and low NTT grades, and/or enabled identification of patients at risk for higher grade toxicities in at least 13 publications [69,128,129,130,131,132,133,134,135,136,137,138]. The main conclusion made in these studies was that the γ-H2AX assay may have a high potential for screening individual radiosensitivity among RT patients. Various methodological aspects of these reports, including radiation doses used, time points investigated, the role of mutations in DNA repair genes, as well as reproducibility and intra-patient variability, are awaiting a specific meta-analysis.

In addition to γ-H2AX, other surrogate markers of DNA DSB repair (e.g., Rad51, BRCA1, 53BP1, pATM, etc.) are specific indicators of different DSB repair pathways that may play a role in the development of NTT [123]. However, none of them have been yet implemented in radiation biodosimetry.

#### 3.4.2. G_2_ Assay

Historically, cytogenetic radiosensitivity tests with irradiation of unstimulated PBL (or other quiescent cells) is called the G_0_ assay, in contrast to the G_2_ assay in which radiation exposure is performed on proliferating cells. The latter method is based on quantification of chromatid-type fragments and is not used for biodosimetry. The reports analysing chromosomal radiosensitivity detected by the G_2_ assay are rather numerous. A rough search in the literature identified 14 papers on the use of G_2_ damage as a marker of genetic predisposition to clinical NTT effects. Seven of these reports contain the conclusion that no direct correlation exists between G_2_ damage and NTT grade [70,75,87,122,126,139,140]. In equal number of studies the opposite result was observed, i.e., cells from patients with severe acute or late NTT effects had a mean G_2_ sensitivity significantly higher than that of the patients without RT-induced normal tissue damage [77,88,91,141,142,143,144]. It should be noted that two positive findings were made using a modification of the G_2_ assay, in which MN were scored instead of chromatid breaks [88,144], and in two studies caffeine was added to the irradiated lymphocyte culture for G_2_-checkpoint abrogation [88,91]. Other known hybrids of the G_2_ approach and DNA damage end-points, like G_2_+γ-H2AX foci (e.g., [145]) or G_2_+PCC (e.g., [146]) have not been reported yet in studies aimed at ex vivo sensitivity related to NTT effects in RT patients.

The G_2_ assay was often used along with the G_0_ test in the same study. In all such reports the authors found that there was no individual correlation between G_2_ and G_0_ damage yields, and each assay identified different patients as radiosensitive [70,75,77,141]. These results suggest that, since different molecular machinery is involved in chromosomal breakage and repair at different stages of the cell cycle, different mechanisms of chromosomal radiosensitivity are likely to operate in G_2_ and G_0_ cells. In general, chromosomally radiosensitive patients may be defective in only one such mechanism, possibly through mutation (or polymorphism) of a single gene. Such mutations may lead to cancer predisposition, of low penetrance, in a large proportion of patients [141]. This hypothesis was supported by a study demonstrating the Mendelian heritability of chromosomal radiosensitivity in family members of breast cancer cases [147,148]. Later, strong evidence for heritability of the G_2_ radiosensitive phenotype was confirmed in another cohort [149].

More information about the possibilities and limitations of the G_2_ assay, covering various aspects of the technique performance, can be found in numerous reports on the use of the G_2_ score as a marker of cancer predisposition. A compilation of data from such studies is beyond the scope of current review. However, one important issue is the intra-individual variations of G_2_ which were investigated in a special study [149]. The heterogeneity was so significant that the authors concluded that too much reliance should not be placed on the result from a single sample when assessing individual radiosensitivity status by the G_2_ assay.

#### 3.4.3. Telomere Length

One more cytogenetic end-point is telomere length. There is some evidence suggesting a link between this parameter and cellular or clinical radiosensitivity. However, the data on the nature of correlation between telomere length and cancer susceptibility (i.e., is the dependence positive or negative?) is rather inconclusive [150]. Moreover, a comparison of telomere length, determined by a flow cytometric FISH assay in PBL of breast cancer patients, failed to reveal differences in cellular radiosensitivity in groups with normal and severe skin reactions to RT [151].

### 3.5. Combination of Biodosimetric Markers with Other Biomarkers of Radiation Response. Multiparametric Approach

The prevalence of a mitotic death pathway for most irradiated normal tissues makes the quantification of ex vivo induced ChA and MN a reliable approach to link the intrinsic radiosensitivity to the NTT in RT patients. However, it should be kept in mind that cytogenetic damage may cover a only a certain range of intrinsic radiosensitivity occurring within a certain range of radiation doses, and may predict not all the types of NTT, but might be best working if the analysis is restricted to specific radiotherapy side effects in patients with one tumor location [152].

On the other hand, it is increasingly accepted that clinical radiosensitivity is likely to be a complex genetic phenotype controlled by genes involved in many cellular processes, including DNA damage recognition and repair, cell proliferation and inter- and intra-tissue signaling. This combination of contributors underlies the inter-individual heterogeneity in radiation effects (damage and repair) in tissues and organs. The genetic determinants of individual radiation susceptibility can be revealed by genomic technologies like mutation detection, SNP analysis or genome-wide association studies. Among prognostic factors, apart from cytogenetic damage (G_0_ ChAs and MN, G_2_ chromatid breaks, γ-H2AX foci), there are a large number of biological end-points, which can serve as a measure of radiation response: DNA breakage and repair, apoptosis, G_2_/M checkpoint arrest, cell survival, colony-forming ability, expression of certain genes, intra- and inter-cell signaling and various biochemical and metabolic changes.

However, for various reasons, the discriminatory power of all known radiation response assays is too low to be used alone in clinical settings, particularly for ex vivo tests. This is not surprising, if one considers that the adverse reactions in patients’ normal tissues may arise from more than one type of underlying defect at cellular level, e.g., the enhanced ChA production may be coupled to the altered apoptosis. Therefore, in clinical practice these biomarkers should not be taken alone, but instead should be included in a compendium of end-points. This is fully applicable to cytogenetic biomarkers. There are many papers presenting the results of ex vivo radiosensitivity assessment using several methods in one study, however, in most such reports, only a simple comparison of prognostic accuracy of different end-points was made. A truly multiparametric approach, where measured effects are combined into an entire prognostic profile, might provide better discrimination.

In the area of interest of this review, there are some examples of such an approach. De Ruyck et al. [143] determined that the G_2_ radiosensitivity assay results, coupled to the risk allele model based on a combination of diverse polymorphisms in DNA repair genes, allowed identification of 23% of the patients with late normal tissue reactions, without false-positive results. In the study of Rzeszowska-Wolny et al. [85], radiosensitive patients were initially selected by a DNA repair test, and then a correlation with NTT in this subgroup was established with the MN assay. Beaton et al. [56] detected a significant increase in the unstable aberration yield in 1st post-radiation mitoses and simultaneously a reduced proportion of cells in 2nd metaphase in ex vivo irradiated lymphocytes of prostate patients, who showed adverse late radiation effects as compared to matched patients exhibiting no adverse effects. In a recent multi-assay study on patients’ fibroblasts, Granzotto et al. [25] showed that the best discrimination among clinically over-reacting patients was provided by the maximal number of pATM foci, and a significant correlation with the NTT severity grade was reached when γ-H2AX foci analysis was added to the results of pATM foci assay, independent of tumor localization and of the early or late nature of the reactions. Further research may help to establish the best combinations of such assays and the “confidence zone” of their application [95].

Among the PBL-based biodosimetry methods currently under development, the most promising are transcriptomics or single gene expression analysis. These technologies have proven to be quite an effective tool for detecting radiation exposure to humans [153,154], including such a complex scenarios as fractionated RT [153,155,156]. There have been several studies that attempted to link ex vivo radiation-induced changes in the expression level of certain genes in patients’ PBLs with their NTT; success in establishing the desired correlation has been regularly reported [87,97,121,157,158,159,160,161,162]. Corresponding changes in gene expression have also been found in RT patients with different grades of NTT in vivo [32,163]. It seems possible that both dose-response markers and NTT predictors can be measured simultaneously within the same transcriptomic platform, providing an ‘all-in-one’ approach with the advantage of full automation and high throughput.

Clearly, more research is needed, in which two or more radiation response biomarkers measured under ex vivo conditions and showing a moderate rate of correlation between them, could be combined using multiple linear regression in order to improve the sensitivity/specificity of prediction of RT-induced NTT.

### 3.6. General Concerns Regarding the Ex Vivo Chromosomal Radiosensitivity as a Predictor of NTT Effects

The intra-individual variability and reproducibility of the cytogenetic assays based on ex vivo irradiation is a very important issue for the radiosensitivity testing and, therefore, requires more comments. In several studies a significant intra-donor variation of radiation-induced cytogenetic damage incidence was found. It was shown that, for the ex vivo assay, the contribution of intra-individual variance to the overall heterogeneity of radiation-induced MN frequencies may be as high as 75% [164]. After both HDR and LDR irradiation regimens, a significant inter-experiment variability was observed in MN yields as well as the dose-rate sparing effect (i.e., reduction in MN yield at LDR compared with HDR) in control donors’ lymphocytes [102]. However, in another study the same researchers noted good reproducibility of the MN assay performed on lymphocytes of 5 normal control donors, whose blood was repeatedly tested 6 times [165]. The conclusions about the ratio of inter-individual to intra-individual variability of cytogenetic radiation response in healthy donors’ cells are contradictory. Some authors showed that the inter-individual variation was significantly higher than intra-individual [166], but other researchers pointed out that there was a high variability between experiments, such that it was not possible to demonstrate inter-individual differences in chromosomal sensitivity. This was true in spite of the use of a control sample from the same normal donor in each experiment [167]. A remarkable, 2-fold increase in variations in radiation-induced cytogenetic damage yields in the same donors’ cells was observed with longer time intervals between repeated samples ranging from 1–3 months to 1 year [81]. The inclusion of the reproducibility test on lymphocytes from healthy control donors has become standard in intrinsic radiosensitivity studies [103,165], but has not changed the overall concern about the results, as the RT patients were only tested once.

Inter- and intra-individual variations of the G_0_ ex vivo MN assay were investigated thoroughly by A. Vral and colleagues [168,169,170]. Repeated experiments on blood cells taken from the same donors over a 1-year period demonstrated that there was no significant difference between intra- and inter-individual variability. Since reproducibility of the assay is determined by the intra-individual variability, these results highlighted the limitations of cytogenetic end points in detecting real, reproducible differences in radiation sensitivity between individuals within a normal population. For example, some healthy donors in the population were identified as being radiosensitive (based on the 90th percentile criterion) but turned out to be normal (non-sensitive) when the assay was repeated at later time points [168,169]. Prolongation of the follow up period up to 3 years did not change the results of testing the repeat samples [170]. The authors stated that the determination of individual radiosensitivity using cytogenetic assays is unreliable when based only on one blood sample, as it may lead to erroneous conclusions. Multiple blood sampling may be necessary to draw reliable conclusions.

There are no reports in the literature presenting a tactics, which can be an alternative to that of suggested by A. Vral et al. [168,169,170] for overcoming the problem of intra-individual variations and low reproducibility of ex vivo radiation cytogenetic assays. As mentioned above, even two radiation doses used in one testing round can produce different classifications for the same individuals [67]. Possibly, building up an entire dose response and further comparison of the curve coefficients or AUCs is a solution for this limitation [80,88].

There are two additional scientific questions, which are somewhat relevant to the problem of the intrinsic chromosomal radiosensitivity. These are (i) the natural general variability of the cytogenetic radiation response in human of lymphocytes, and (ii) the specific traits of chromosomal radiosensitivity in cancer patients versus healthy donors. A large number of reports can be found in the literature on each of these questions, but these studies did not register clinical NTT effects, therefore their results have limited value for radiation oncology. Their detailed analysis is beyond the scope of the current work; however, these issues will be highlighted in the forthcoming IAEA Health Series Report [38] in relevance to other clinical applications of cytogenetic biodosimetry. Actually, in the correctly executed NTT studies a possible impact of the aforementioned factors can be minimized by (i) the presence of a sufficient number of cases in the study, and (ii) the inclusion of positive controls, i.e., patients without radiation lesions, and negative controls, i.e., unirradiated healthy donors.

As was mentioned at the beginning of this review, the key idea of the prediction of the NTT by ex vivo tests is that elevated chromosomal radiosensitivity and a predisposition to the abnormal NTT response to RT are both attributable to patients’ genetics. Therefore, it is very tempting to assume the presence of a mechanistic link between these two traits. Data obtained by Widel et al. [81] best supports this assumption: In lymphocytes irradiated ex vivo, the mean yield of MN was significantly higher in samples from patients demonstrating acute and/or late normal tissue reactions, than in those from patients showing no reactions; however, healthy donors fell between the two patient groups. This may suggest that the control healthy donors group may contain both radiosensitive and radioresistant individuals, and that some of them may be potential clinical over-reactors. Therefore, it should be recommended that a matched (or at least, large enough) group of healthy donors always be included in the ex vivo radiosensitivity testing in order to guarantee the quality control of the studied population.

Also the aspect of the patient’s age might play a very important, dual role. First, aging tissues might intrinsically harbor more DNA damage that could sensitize (or not) to RT, thus modulating the NTT occurrence. Second, there are serious concerns about the equality of the cytogenetic dose response (i.e., chromosomal radiosensitivity) in cells of young vs. old donors [171,172]. To the best of our knowledge, no one research group specifically considered the age factor in their studies on ex vivo cytogenetic tests for the NTT prediction. This might be a task for future research.

Also, it is important to determine the best method for the initial stratification of patient cohorts for data analysis: either according to the chromosomal radiosensitivity or clinical response. If the former is chosen as a discriminator, then the shape of the ChA or MN frequency distribution within a cohort should be thoroughly analysed, and the cut-off criteria must be clearly defined. The most frequent approach is to check the observed distribution for consistency with Gaussian statistics and to carry out a classification based on the arbitrarily chosen definitions ≤MV–SD as resistant, MV ± SD as normal and ≥MV + SD as sensitive, where MV is mean value and SD is standard deviation of the mean. It is apparent, that such a classification does not consider the normal probability for any individual in the group to be located in any of three categories after a single sampling. Therefore, “two doses, two times” can be recommended as a minimum experimental design for unbiased assigning of a patient to a certain category of cytogenetic radiosensitivity. However, it is not yet clear whether the definition based on MV and SD is applicable and how it should be modified, if the test is performed two or more times, or is based on two or more radiation dose points.

If it is possible to create a full ex vivo dose response curve for each patient in the study, then the efficacy of the AUC versus curve coefficients ± error as a discriminator have to be evaluated.

Irrespective of study design, it should be kept in mind that the difference in radiation-induced aberration yield per unit radiation dose between individuals can be rather small. Therefore, in order to validate the suggested assays, QA/QC actions aimed at strengthening the reproducibility should be supplemented with normalization of the individual data using internal standards, as was suggested earlier for clonogenic end-points of cellular radiosensitivity, which also suffer from intra-individual variation [173,174].

Another aspect of the problem is the evaluation of NTT per se. Focusing the study on one type/location of normal tissue damage (e.g., skin) reduces uncertainties compared to the inclusion of various NTT effects graded by a certain scoring system. Also it is plausible that different cytogenetic assays could identify different response phenotypes associated with acute or late reactions [77].

A prospective study design seems to be the best for the development of prognostic test, as it avoids the uncertainty caused by RT induced ChA or MN yields. For the analysis of late NTT effects, patients should be surveyed long enough after RT to cover the latency period for clinical effects.

If in the radiosensitivity study the cytogenetic data are used as the primary factor for patient group stratification, and the NTT effect is a dependent parameter, then the latter should be assessed for the grade, the frequency and the on-set time. Thus, the most comprehensive approach for clinical practice includes the stratification of patients according to the results of the comparison of AUCs of their individual ex vivo dose responses, followed by generating the predictive risk-analysis actuarial curves for complication-free survival for a given grade of the certain NTT effect.

Recently a method was suggested, by which the patients were identified on the basis of moderate/marked or minimal/no NTT adverse effect despite the absence or presence of variables predisposing the patient to this particular effect [137]. Risk factors for adverse RT effects can then be established by multivariate analysis of the NTT outcomes. For example, in that report the favourable factors (lower NTT risk) in breast cancer patients were the lower whole breast RT dose, 3D dosimetry, no boost dose to the tumor bed, small breast size, minimal surgical cavity and no axillary RT. Patients with striking adverse effects despite favourable parameters were classified as ‘RT-Sensitive’, and unmatched patients with no changes even with unfavourable parameters were considered as ‘RT-Resistant’. This approach allows maximum separation in terms of intrinsic factors predisposing the patient to the presence or absence of adverse NTT effects. In this report a significant association between the NTT effects and ex vivo γ-H2AX foci yields was established particularly in lymphocytes, whereas no such correlations was observed in cultured skin cells (fibroblasts, endothelium, keratinocytes and epidermis [137]. However, to the best of our knowledge, such a classifier based on the ‘despite-predisposing-variables’ principle, has not yet been applied in NTT-radiosensitivity studies using biodosimetric markers. Surely, more validation studies on the reliability of such an approach are required.

Other aspects of radiation biomarker research in relation to clinical radiosensitivity, including the underlying rationale, the necessity formeticulous recruitment of patients, study design that accounts for clinical factors, which modify normal tissue responses, as well as some limitations and confounding factors that affect tests of association between predictive markers and clinical radiosensitivity, have been highlighted in reviews [16,17,18,19,20,21,22,23,31,32,95].

## 4. Conclusions and Recommendations

In RT, normal tissue reactions are often the regulating factor for treatment. As such, there is no robust screening method to predict normal tissue reactions to RT, particularly in comparison to tumor tissue. Such a screening method would allow radiation dose to be tailored to each patient. On the basis of numerous studies, it is reasonable to conclude that the severity of RT-related complications is essentially determined by genetic predisposition, which can be revealed and quantified in normal cells. Human PBL are the preferred tissue for assays of NTT response (particularly, as an alternative to fibroblasts) due to the ease of obtaining samples and the rapid generation of the results. In cancer patients, evidence suggests that enhanced PBL radiosensitivity, assessed by various end-points, associates with the development of RT-related morbidity. Therefore, the attempts to develop clinically applicable tests based on radiation cyto- or genotoxicity in lymphocytes as a rapid predictive biomarker of normal tissue radiosensitivity are convincing and logical.

ChA frequency is considered a good indicator, because cytogenetic damage is usually related to an altered DNA repair function, which is in turn linked to cellular radiosensitivity, for which a dysfunction of many elements of DNA damage sensing and repair have been demonstrated. This has been strongly supported by the clear success of cytogenetic analysis of cases with inherited DNA repair defects, identified by molecular or clinical signs, which are always confirmed by abnormal results of post-ex vivo irradiation cytogenetic analysis.

However, for the rest of the over-reacting patients, the results appear to be rather controversial. In approximately 50% of the reports, the average yield of biodosimetric markers was higher in over-reacting patients than in patients with lower grade NTT. Also, a significant correlation was sometimes found between the biomarker yield and the severity of acute or late NTT reactions at an individual level, but this observation was not unequivocally proven. Both the presence and the absence of correlations between cytogenetic damage frequency and acute or late normal tissue effects after RT were reported by the same and by different research teams. Thus, it is possible that, for different cytogenetic radiosensitivity phenotypes, their associations with NTT effect might be irradiation site- and damaged organ/tissue-specific.

The inter-individual variations of ex vivo ChA or MN yields in over-reacting patients is similar to or wider than that of patients without adverse NTT effects. In the majority of studies the overlap between the distributions of individual frequencies of cytogenetic damage in cells taken from patients with high-grade and low-grade NTT reactions did not allow clear identification of persons at risk by an ex vivo test. That is one of the main reasons for the limited application of biodosimetric markers for identifying radiosensitive individuals among RT patients. The second reason is the intra-individual heterogeneity, which determines the reproducibility of the assay, and which has not been studied thoroughly enough in RT cohorts. Instead, there is a serious concern, coming from cancer risk studies, that the determination of individual radiosensitivity with cytogenetic assays is unreliable when based on a single measurement, and multiple blood sampling is necessary to get reliable patient classification.

Thus, a general conclusion is that the assays based on ex vivo biodosimetric markers in PBL in their present form are unlikely to result in the development of a reliable ‘stand-alone’ assay of radiosensitivity, which can be of assistance for the prediction of NTT effects in the clinic and lead to individualized patient RT schedules. The following are some suggestions how these issues can be addressed:Patient groups, selected for prospective studies, should be large enough to provide a sufficient number of cases of adverse NTT. In retrospective studies, a case-control design is preferable with well-matched control patients. A healthy donors group should also be included in the study.The formation of “teaching” datasets for the primary search for a correlation between ex vivo induced biomarker yield and the NTT should be undertaken through stratifying the patients according to their clinical effects. A “despite-predisposing-variables” approach [137] should be used, where possible, to guarantee the maximum separation of clinically radiosensitive and radioresistant patients in terms of intrinsic factors predisposing to the presence or absence of adverse NTT effects.It is highly desirable to maintain the second means of patient stratification according to molecular classification of human radiosensitivity [25]. Respective predictive assays should be performed to separate the radioresistance group; the group of moderate radiosensitivity caused by delay of nucleoshuttling of ATM (includes majority over-reacting patients), and the group of hyper-radiosensitivity caused by a gross DSB repair defect. The biodosimetric markers may be applied for further partition of the over-responding patient group.A set of criteria of excellence for these types of study should be maintained:-minimum confounders, i.e., one tumor site, one irradiation scheme and irradiated sites locations;-one type of NTT (one organ or tissue) per study; NTT grade, frequency and latency assessed;-ChAs are more preferable than MN;-at least 2 radiation doses ex vivo (the higher of two doses has to be AHAP), at least 2 repeats of the assay for each individual in the studied cohort;-alternatively, a full dose response should be built for each individual according to classical biodosimetric methodology (minimum 6 dose response points to estimate 3 coefficients of the classic linear quadratic model); the result is the set of coefficients with their errors or the entire AUC.The “teaching” phase should be finalized by generating prognostic risk-analysis actuarial curves for complication-free survival (frequency and latency time) for various grades of the studied NTT effect.In the validation phase of the ex vivo biomarker study, its predictive efficacy should be assessed by a common test for general accuracy (sensitivity/specificity), and re-evaluated by stratifying patients according to their intrinsic cytogenetic radiosensitivity and calculating the annual risk for a given grade of the NTT effect using the actuarial curves.The use of internal standards for the determination of the intrinsic radiosensitivity in patients’ lymphocytes at each stage of the research should aid the development and evaluation of the prognostic tests.

To make cytogenetic ex vivo irradiation-based assays more attractive for clinical applications, they can be combined with automated scoring of cytogenetic damage using flow cytometry or computerized image analysis systems [175,176].

These recommendations may help to develop the ex vivo tests, which would be feasible in clinical practice and could be used as supplementary markers in radiobiological control for radiation oncology. To accomplish this, more retrospective, case-control studies are needed, along with larger prospective studies to confirm existing findings. This will help validate the use of ex vivo cytogenetic assays in the future to predict normal tissue radiosensitivity and discriminate individuals with marked early and late normal tissue reactions after RT. A coordinated approach among different laboratories would be useful to set the relevant standards and increase sample numbers to allow for robust analysis and strong conclusions that will help convince the radiation oncology community to adopt these predictive assays.

## Figures and Tables

**Table 1 jpm-10-00285-t001:** Ex vivo tests for normal tissue toxicity (NTT) prediction: chromosome aberrations in human blood lymphocytes.

Reference	Patients and Study Type	Test System and Ex Vivo Exposure Details	Normal Tissue Toxicity	Correlation
**Dicentrics and fragments—conventional analysis in metaphases or prematurely condensed chromosome (PCC)spreads**				
Jones et al., 1995 [63]	Retrospective; 16 breast cancer patients. Exaggerated acute or late radiation reaction of normal tissues after radiotherapy RT; no positive control (i.e., matched RT patients without acute or late reactions)	LDR ^1^ (0.0031 Gy min^−1^) and HDR ^2^ (0.17 Gy min^−1^) irradiation to 3 Gy γ-rays	Early reactions: erythema, moist desquamation. Late complications: fibrosis, telangiectasia.	Abnormal chromosomal radiosensitivity was found in 5 of 7 patients with excessive early skin reactions. The mean ChA ^3^ yield after LDR for early over-reactions was significantly higher than for healthy controls and average sparing was less. LDR-induced yields were above the control range in 2 out of 10 patients with late complications. The mean yield for late over-reactors was not significantly above that of controls. Also, one early overreactor and one late overreactor had LDR aberration yields below the control range.
Kondrashova et al., 1997 [64]	Retrospective; 12 patients with different cancers, all with late radiation skin damage, studied 0.4–31 years after RT (no positive control, i.e., matched RT patients without acute or late reactions)	Acute irradiation (0.2 Gy min^−1^) to 2 Gy γ-rays	Late radiation skin injuries (grade not specified)	In 3 out of 12 patients the frequency of ex vivo induced chromosome type fragments significantly exceeded the control value.
Borgmann et al., 2002 [65]	Retrospective; 16 pair-wise matched head and neck cancer patients, exhibiting maximum differences (8 grade 1 vs. 8 grade 3) in late normal tissue reactions 2–7 years after RT.	Acute irradiation (2 Gy min^−1^) to 2, 4 and 6 Gy X-rays. Conventional dicentrics and ‘fusion’ PCC methods.	Fibrosis, telangiectasia, mucositis and xerostomia assessed using the RTOG ^4^ score	At 6 Gy ex vivo irradiation the mean yield of aberrations and PCC fragments in PBL ^5^ of overreacting patients was significantly higher than in cells from patients with mild reactions and healthy controls. The pair-wise match of patients revealed that in all except one case the grade 1 individual had less ex vivo aberrations than the grade 3 counterpart.
Hoeller et al., 2003 [66]	Retrospective; 86 breast cancer patients with or without late fibrosis 5–17 years after RT.	Acute irradiation (2 Gy min^−1^) to 6 Gy X-rays	Fibrosis, LENT-SOMA ^6^ score, grades 0, 1, 2 or 3	Patients with high cellular radiosensitivity (ex vivo yield > mean + 1 standard deviation) showed a higher annual rate forfibrosisthan patients with low or intermediate radiosensitivity (3.6% versus 1.6% per year).
Borgmann et al., 2008 [67]	Prospective; (A) 51 patients with different tumor sites, and (B) 87 breast cancer patients.	Acute irradiation (2 Gy min^−1^) to 3 or 6 Gy X-rays. Culturing for 72 h with noBrdU ^7^ control!	Acute reactions assessed using the RTOG score	The fraction of patients with Grade 2–3 reaction increased with increasing individual radiosensitivity, measured by the yield of chromosome fragments at 6 Gy.
Tang et al., 2008 [68]	Retrospective; pair-wise matched nasopharyngeal carcinoma patients with (26 persons) or without (26 persons) radiation encephalo-pathy	Acute irradiation (2 Gy min^−1^) to 6 Gy photons (6 MeV linear accelerator)	Radiation encephalopathy assessed using the RTOG score	The mean aberration yield ex vivo in patients with Grade 3–4 reaction was higher than that in patients with Grade 1–2 reactions and controls. Patients with high cellularradiosensitivity (ex vivo yield > mean + 1 standard deviation) showed shorter latency for the encephalopathy development compared to those with a low or intermediate radiosensitivity.
Chua et al., 2011 [69]	Retrospective; 14 pair-wise matched breast cancer patients (7 cases with late radiation skin damage and 7 controls with no damage)	Acute irradiation (0.5 Gy min^−1^) to 6 Gy X-rays. Culturing for 72 h with no BrdU control!	Scores of severe radiation-induced change (cases) or very little/no change (controls) in the breast on photos taken before RT and at 2 and 5 years post-RT	In 5 out of 7 clinically radiosensitive cases the total yield of aberrations ex vivo remarkably exceeded its top level in the matched control group. The mean yields of dicentrics and excess acentrics ex vivo were statistically higher in cases than in controls.
Padjas et al., 2012 [70]	Prospective; 35 patients with breast cancer and 34 with gynaecological cancer	Acute irradiation (2 Gy min^−1^) to 2 Gy photons (6 MeV linear accelerator)	Early and late side effects assessed using the RTOG score	No correlation was observed between the results of the cellular radiosensitivity assay and the severity of side effects.
Beaton et al., 2013 [56]	Retrospective; 10 prostate cancer patients with grade 3 late radiation proctitis and 20 matched patients with grade 0 proctitis.	Acute irradiation (1.7 Gy min^−1^) to 6 Gy X-rays. Culturing for 72 h with BrdU control.	Late proctitis assessed using the RTOG score	The mean yields of dicentrics and excess acentric fragments were statistically higher in clinically radiosensitive patients than in the control group. A sufficient proportion of Grade 3 patients showed the induced acentric yield above the upper limit for this end-point observed in Grade 0 group.
**FISH-detectable breaks, Tn and/or CCR in metaphases or PCC spreads**				
Dunst et al., 1995 [60]	Retrospective; 16 patients (12 breast cancer and 4 other cancers), including 4 persons with increased clinical radiosensitivity and 12 with normal tolerance to RT, examined 1 to 108 months after treatment.	Acute irradiation to 0.7 or 2 Gy X-rays from 6 MeV linear accelerator. Radiation-induced breaks per mitoses assessed by FISH/CISS ^8^ technique	1 severe acute reaction in bladder; 1 acute skin reaction with subsequent fibrosis of breast; 1 radiation myelitis; 1 severe acute reaction after mediastinal irradiation	4 patients with increased clinical radiosensitivity showed statistically increased chromosomal radiation-induced damage as compared to the 12 patients with normal radiation tolerance at both ex vivo radiation doses.
Neubauer et al., 1997 [71]	Prospective group: 33 breast cancer patients; retrospective group: 28 breast cancer patients and 19 other tumor locations. In total 66 patients (some investigated before and after RT)	Acute irradiation (2.2 Gy min^−1^) to 0.7 or 2 Gy X-rays from 6 MeV linear accelerator. Radiation-induced breaks and CCR per mitoses assessed by FISH/CISS technique	Acute effects assessed using the WHO ^9^ grading system and late side effects assessed using the RTOG score	The proportion of breaks, involved in CCR, after 0.7 Gy ex vivo, was remarkably higher in 27 samples patients with high toxic reactions, compared with 20 samples from patients with average clinical reactions and 19 healthy controls. The yield of mitoses with CCR was increased proportionally to the clinical reactivity at both ex vivo radiation doses, but the tendency was especially pronounced at 2 Gy.
Dunst et al., 1998 [72]	Prospective group: 26 patients; retrospective group 26 patients. In total 52 patients: 41 with breast cancer, 11—other sites (lung, head and neck prostate, bladder, rectal cancer and Hodgkin’s disease).	Acute irradiation to 0.7 or 2 Gy X-rays (6 MeV linear accelerator). Radiation-induced breaks per mitoses assessed by FISH/CISS technique	Acute effects assessed using the WHO grading system and late side effects assessed using the RTOG score	A significantly higher number of chromosomal breaks were found after both radiation doses ex vivo in lymphocytes of 9 patients with severe or extreme radiation reaction of normal tissues as compared to 43 patients with no or mild to moderate radiation reactions.
Keller et al., 2004 [73]	Retrospective group: 5 patients with severe radiation-induced late effects of Grade ≥3, 18–76 months after RT for cancers of different locations, versus 11 healthy individuals; no positive control, (i.e., matched RT patients without late reactions).	Acute irradiation (2.2 Gy min^−1^) to 0.7 or 2 Gy X-rays from 6 MeV linear accelerator. Radiation- induced FISH-detectable breaks and CCR, dicentrics, translocations, excess acentrics per mitoses assessed by FISH/CISS technique	Late effects assessed using the RTOG score	The ratio of the mean yields in radiosensitive patients to that of in healthy donors after 2 Gy ex vivo varied from 1.2 to 1.8 for breaks, translocations, dicentrics and excess acentrics, and increased to 3.2 for CCRs. The “frequency of breaks per metaphase”, “CCR per metaphase” and “translocations per metaphase” appeared to be the most suitable parameters to detect a difference in chromosomal sensitivity between healthy and clinically radiosensitive individuals.
Huber et al., 2011 [74]	Prospective group: 47 breast cancer patients. Acute skin reactions: 4 patients showed grade 0, 30 patients grade 1, 12 patients grade 2, and 1 patient grade 3.	Acute irradiation (0.5 Gy min^−1^) to 3 Gy X-rays. Radiation-induced FISH-detectable dicentrics, Tn and colour junctions assessed by FISH technique (Dic were counted only in painted chromosomes)	Acute effects scored according to the NCI-CTC ^10^ scale. Also according to the time of the skin reaction occurrence, patients were divided into “early”, “in between reaction” and “late reaction”.	Three out of 4 patients with increased chromosomal radiosensitivity showed either more severe side effects (1 patient) or early onset of the skin reactions (2 patients). A significant overall correlation was found between the ex vivo frequencies of Tn and the latency of side effects of the skin. With a definite cut-off for Tn yield, 22 of the 30 short latency patients were correctly detected (73.3% sensitivity) and 11 of the 17 longer latency patients (were correctly assigned (64.7% specificity).
Beaton et al., 2013 [57]	Retrospective group: 10 prostate cancer patients with Grade 3 late proctitis versus matched 10 prostate cancer patients with no proctitis.	Acute irradiation (1.7 Gy min^−1^) to 4 Gy X-rays. Radiation-induced, FISH-detectable ChA and colour junctions	Late proctitis assessed using RTOG/EORTC Late Toxicity Scale	After 4 Gy ex vivo irradiation, the clinically radiosensitive group had significantly higher rates of chromosome damage in the number of colour junctions per cell, the number of deletions per cell and the number of dicentrics per cell, compared to proctitis-free control.
Schmitz et al., 2013 [75]	Retrospective group: 10 prostate cancer patients with acute (0 or 2 months after RT) or late side effect (16 months after RT) versus 10 patients without severe side effects versus 11 healthy controls	Acute irradiation (0.74 Gy min^−1^) to 0.5, 1.0 and 2.0 Gy ^137^Cs γ-rays. Culturing for 72 h with no BrdU control! Radiation-induced FISH-detectable dicentrics, Tn, centric rings, excess acentrics per mitoses assessed by FISH technique	Early and late severe side effects assessed with the validated Expanded Prostate Cancer Index Composite questionnaire (EPIC)	Prostate cancer patients with and without side effects cannot be distinguished from healthy donors based on the mean aberration yield after ex vivo irradiation. The distribution pattern of the aberrations per donor did not differ in each donor group after exposure to any dose ex vivo.

^1^ LDR—Low dose-rate. ^2^ HDR—High dose-rate. ^3^ ChA—chromosome aberrations. ^4^ RTOG (also RTOG/EORTC)—The Radiation Therapy Oncology Group and the European Organization for Research and Treatment of Cancer criteria. ^5^ PBL—peripheral blood lymphocytes. ^6^ LENT-SOMA—The Late Effects in Normal Tissue—Subjective, Objective, Management, Analytic criteria. ^7^ BrdU—Bromodeoxiuridine, a reagent typically used in radiation cytogenetics for the control of the number of cell cycles passed by a particular cell in culture [39].^8^ CISS—Chromosomal in Situ Suppression Hybridization technique. ^9^ WHO—World Health Organization. ^10^ NCI-CTC—Common Toxicity Criteria of the United States National Cancer Institute.

**Table 2 jpm-10-00285-t002:** Ex vivo testsfor NTT prognoses: micronuclei in human blood lymphocytes as the end-point.

Reference	Patients and Study Type	Test System and Ex Vivo Exposure Details	Normal Tissue Toxicity	Correlation
Rached et al., 1998 [76]	Retrospective group: 15 patients with various cancers experiencing severe acute reaction of normal tissue, 15 non-matched cancer patients without reactions and 15 healthy donors.	Acute irradiation (1.08 Gy min^−1^) to 4 Gy X-rays.	Mucositis, diarrhea, epitheliolysis, proctitis	There was no difference between cancer patients with or without acute reactions in normal tissues in their MN scores after ex vivo irradiation.
Barber et al., 2000 [77]	Breast cancer patients. Prospective group: 123 patients studied before RT, 116 tested with the HDR assay, 73 with the LDR assay. Retrospective group: 8–14 years after RT, 47 tested with the HDR assay, 26 with the LDR assay.	HDR assay: Acute irradiation (3.0 Gy min^−1^) to 3.5 Gy ^137^Cs γ-rays. LDR assay: protracted irradiation (dose rate 0.15 cGy min^−1^, total exposure time 38.8 h) to 3.5 Gy ^137^Cs γ-rays. Throughout the LDR irradiation period the samples were maintained at 37 °C in 5% CO_2_ atmosphere. Culturing for 90 h!	Acute skin reactions scored as minimum erythema, moderate erythema or severe erythema/moist desquamation/edema. Late effects assessed according to the LENT-SOMA	In the prospective group with and without acute reactions there was no significant difference between clinically hyper-sensitive (HR) and non-HR patients for the MN yield induced ex vivo either at HDR or LDR. Regarding late effects: mean HDR and LDR MN scores were higher in 4 patients with severe telangiectasia than in those with normal reactions and 8 patients with severe fibrosis had higher HDR MN scores than the normal reactors. However, the HDR assay’s sensitivity for detecting HR cases was 0/6 for acute reactions, 4/8 for late fibrosis, 2/9 for breast retraction and 2/4 for telangiectasia. For LDR assay’s sensitivity that was 0/2, 0/5, 0/3 and 2/4, respectively.
Słonina et al., 2000 [78]	Prospective group: 12 cervical cancer patients and 11 head and neck cancer patients. Retrospective group: 9 cervical cancer patients and 1 head and neck cancer patient, all late reactors (4–14 years after RT).	Acute irradiation (0.73 Gy min^−1^) to 0, 2.0 and 4.0 Gy ^60^Co γ-rays.	Acute and late reactions were assessed according to RTOG/EORTC grading system for 8–50 months in prospective group.	There was no correlation between the radiosensitivity, assessed as induced ex vivo MN yield, and acute or late clinically observed side effects in RT patients.
Lee et al., 2000 [79]	Prospective group: 8 prostate cancer patients. Blood taken before RT and during RT.	Before RT: Acute irradiation (0.8 Gy min^−1^) to 0, 1.0, 2.0, 3.0 and 4.0 Gy ^137^Cs γ-rays. During RT: Acute irradiation (0.8 Gy min^−1^) to 0 and 2.0 ^137^Cs γ-rays.	Acute side effects: cystitis, diarrhea	In 2 of 3 patients with Grade I RT-induced early side effects the MN yield in PBL induced by ex vivo irradiation before RT was significantly higher than in the other patients without RT-induced side effects. For the second blood samples obtained during the second half of RT course the MN yields in PBL induced by 2 Gy ex vivo irradiation had no predictive value.
Lee et al., 2003 [80]	Prospective group: 38 prostate cancer patients: over-reactors (OR, 13 patients with Grade ≥2 RT-related morbidity) and average reactors (AR, 25 patients with Grade 0–1 RT-related morbidity). Strict patient selection criteria.	Acute irradiation (0.9 Gy min^−1^) to 0, 1.0, 2.0, 3.0 and 4.0 Gy ^137^Cs γ-rays.	Gastrointestinal (GI) and genitourinary (GU) morbidity assessed according to the RTOG criteria.	The averaged dose response for ex vivo induced MN was remarkably more intensive in OR than in AR; the differences in MN yield were highly significant at doses ≥2 Gy. Also, for both AR and OR groupss, the inter-individual variation of the ex vivo dose response of MN yields was greater than that of the intra-individual variation.
Widel et al., 2003 [81]	Prospective group: 55 cervical carcinoma patients. Verified group due to strict patient selection criteria. Control group—25 healthy female donors.	Acute irradiation (0.8–1.0 Gy min^−1^) to 0, 2.0 and 4.0 Gy ^60^Co γ-rays.	Acute reactions during RT and up to 3 months after RT assessed by the Common Toxicity Criteria of the National Cancer Institute and RTOG. The late effects were classified according to the RTOG/ EORTC grading system	In lymphocytes irradiated ex vivo to 4 Gy the mean yield of MN was significantly higher in samples from patients, who suffered from acute and/or late normal tissue reactions, than in those from patients with no reactions, but in healthy donors the value fit in between two patients groups. A significant correlation was found between individual MN yield at 4 Gy and the severity of acute reactions and late reactions. However, the essential overlap between the distributions of individual MN frequencies in patients with high-grade and low-grade reactions does not clearly allow identification of persons at risk by MN test.
Bustos et al., 2002 [82]; Di Giorgio et al., 2004 [83]	Retrospective group: 19 head andneck cancer patients 6–18 months after RT	Acute irradiation to 0 and 2.0 Gy ^60^Co γ-rays.	“Late” reactions to RT: osteonecrosis, fibrosis and trismus	In 3 out of 4 patients, who had developed late reactions, the ex vivo induced MN yield was significantly higher than in lymphocytes from the rest of the patients. The individual cytogenetic response ex vivo showed a correlation with the maximum grade of late reactions.
Taghavi-Dehaghani et al., 2005 [84]	Retrospective group: 26 breast cancer patients, including 15 with acute reactions and 11 with late reactions (no positive control, i.e., matched RT patients without acute or late reactions). Time after RT not specified.	Acute irradiation to 0 and 4.0 Gy ^60^Co γ-rays.	Early tissue damage: erythema, dry desquamation, moist desquamation. Late tissue damage: fibrosis, skin telangiectasia, pigmentation.	The mean yield of MN after 4 Gy ex vivo was significantly (1.6 times) higher in lymphocytes of patients with early reactions, than that of patients with late reactions.
Rzeszowska-Wolny et al., 2008 [85]	Prospective group: 34 head andneck cancer patients.	Acute irradiation (1.14 Gy min^−1^) to 0, 2.0 and 4.0 Gy ^60^Co γ-rays.	Acute reactions measured using the Dische scale	In a subgroup of 14 patients with a high level of induced residual DNA damage, measured using a ‘comet’ assay ex vivo, a statistical correlation occurred between the MN yield after 4 Gy ex vivoandacutetoxicity score.
Encheva et al., 2011 [86]	Prospective group: 23 cervical cancer and 17 endometrial cancer patients.	Acute irradiation (1.0 Gy min^−1^) to 0 and 1.5 Gy ^60^Co γ-rays.	Acute normal tissue reactions were graded according to the NCI-CTC for Adverse Events v.3.0.	Great variations in MN yield ex vivo were found, but no correlation occurred between cytogenetic effect and clinical radiosensitivity. The resultant conclusion is against a recommendation of ex vivo MN test for clinical use.
Finnon et al., 2012 [87]	Retrospective group: breast cancer patients with marked (31 cases) or mild (28 controls) late adverse reaction to adjuvant breast RT	Acute irradiation to 3.5 Gy X-rays. Cell culturing for 90 h!	Scores of severe radiation-induced change (cases) or very little/no change (controls) in the breast on photos taken before and after RT.	Significant inter-individual variations in radiation- induced MN were observed, but there was no evidence of a differential response in cases and controls in matched or unmatched analyses, e.g., variations in cytogenetic ex vivo radiosensitivity did not correlate with normal tissue response to RT.
Vandevoorde et al., 2016 [88]	Retrospective group: 12 breast cancer patients expressing severe radiation toxicity matched to 12 controls with no or minimal reactions, with a follow-up for at least 3 years	Acute irradiation (0.14 Gy min^−1^) to 5 doses from 0.2 to 3.0 Gy ^60^Co γ-rays.	Late adverse reactions assessed by LENT-SOMA scale and by comparing standardized photographs pre- and post-RT resulting in an overall cosmetic score.	The average dose response curve of the ex vivo induced MN yield for cases lies significantly above the average dose response curve of the controls, and the coefficients of the LQ dose response do not overlap between cases and controls. However the shift in the dose response from case to control on an individual basis was not systematic, indicating no direct correlation of the MN induction with the clinical radiotoxic effects.
Batar et al., 2016 [89]	Prospective group: 40 [89] and later on 100 [90] breast cancer patients, including 20 [89] or 50 [90] ‘cases’ with acute reactions (grades 2, 3 or 4) and 20 [89] or 50 [90] ‘controls’ with no or mild adverse.	Acute irradiation (1.0 Gy min^−1^) to 0 and 2.0 Gy ^60^Co γ-rays.	Acute normal tissue reactions were followed during 6 weeks after RT and graded using CTC: Grade: 0 (no adverse effect), 1 (mild adverse effect), 2 (moderate adverse effect), 3 (severe side effect) and 4 (life-threatening adverse effect).	The MN yield was higher in the group with acute reactions (2.10 ± 0.27) than in control patients (1.67 ± 0.20), but the difference was not statistically significant.
Batar et al., 2018 [90]				There was no difference in the mean MN frequency between the group with acute reactions (6.8 ± 4.2) and controls (6.9 ± 2.6).
Guogytė et al., 2017 [91]	Prospective group: 4 prostate cancer patients and 1 uterine cancer patient.	Acute irradiation to 0 and 2.0 Gy X-rays.	Acute normal tissue reactions, including GI and GU were graded according to the RTOG/EORTC.	The ex vivo MN yield in 2 patients with Grade 1 side effects increases by 8% compared with that in 2 patients without side effects. The patient with Grade 2 side effects had the ex vivo MN yield 9% higher than that in Grade 0 case and 1% higher than in patients with Grade 1 side effects.
da Silva et al., 2020 [92]	Prospective group: 10 cervical cancer patients, including 3 patients treated with teletherapy alone and 7 receiving teletherapy and brachytherapy.	Acute irradiation (2 Gy min^−1^) to 2.0 Gy X-rays on a linear accelerator.	Acute normal tissue reactions, that were developed in patients 5–10 days after starting the radiation treatment, were graded according to the RTOG.	The ex vivo MN yield showed a significant correlation with the RTOG score (*r* = 0.96). However, the baseline MN yield in non-irradiated cultures also had a significant correlation with the severity of adverse effects (*r* = 0.86). The re-analysis of the original data showed a strong association between MN yields in ex vivo irradiated and sham irradiated samples (*r* = 0.90) and very moderate correlation of the truly induced MN yield (the difference between irradiated and non-irradiated samples) with the toxicity score (*r* = 0.44). Actually, only 1 out of 3 patients with severe NTT can be identified confidently by the ex vivo induced MN yield.
Chaouni et al., 2020 [93]	Retrospective group: 18 patients with Merkel Cell Carcinomas, including 9 patients with Grade ≤2 and 9 patients with Grade ≥3 NTT in skin. Exact time after RT not specified.	Acute irradiation (2 Gy min^−1^) to 2.0 Gy and 10.0 Gy photons on 6 MeV linear accelerator.	Late skin reactions assessed according to RTOG/EORTC grading system.	Inverse correlation between the ex vivo induced MN yield and NTT. The difference between 0 Gy and 2.0 Gy points were 333 MN per 1000 BN in Grade ≤2 group and 218 MN per 1000 BN in Grade ≥3 group; between 0 Gy and 10.0 Gy that, respectively, were 2663 MN per 1000 BN and 854 MN per 1000 BN. The 3.1-fold difference between groups with Grades ≤2 and ≥3 NTT for the MN yield after 10 Gy was statistically significant.

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
