# Peer review of "Prediction of the Acute or Late Radiation Toxicity Effects in Radiotherapy Patients Using Ex Vivo Induced Biodosimetric Markers: A Review"

_jpm, 2020, doi:10.3390/jpm10040285_

Round 1

Reviewer 1 Report

The authors have provided a quite extensive overview on the “past, present and future” of “ex vivo” testing for NTT evaluation. After an introduction of markers used for cytogenetic biodosimetry, case studies are critically discussed and sorted by central aspects: efficiency of diverse end point analysis, dose dependency, statistical testing, using diverse cell types and also focusing on other DNA damage markers. At the end the manuscripts concludes on substantial limitation on these at the present used assays and tries to provide solutions on the “How to predict better” by making correct use of these cytological markers. The idea that such “ex vivo” cytological markers are a perfect tool, to estimate how the cells genetic repertoire is equipped to radiation damage, is pushed forward throughout the manuscript.

Still some minor clarification and addition could be beneficial for the manuscript.

Before extensively discussing efficiency of cytological markers it would be helpful to clarify the “mode of action” of radiotherapy (how does radiotherapy exactly kills cancer cells). How specific is it to cancer cells and is there any relationship to pre-existing DNA damage in cancer cells (for example in chromosomal instable cancer cells) and RT efficiency?

The aspect of tissue specificity is just briefly mentioned in the manuscript and yet it could be very important aspect considering tissue toxicity. We easily can imagine that not all cell types are equally equipped to respond to DNA damage.

Also the aspect of the patient’s age might play a very important role as ageing tissue might intrinsically harbor more DNA damage that could sensitize (or not) to radiation therapy.   

Reviewer 2 Report

In this manuscript by Vinnikov et al., authors have performed a thorough analysis of the available data on the predictive power of cytogenetic assays from biodosimetry as to the occurrence of normal tissue toxicities during radiotherapy. Acute and chronic side effects of radiotherapy can significantly impair the patient's quality of life and are a limiting factor of cumulative radiation dose and tumor control. This work is of interest and relevance for clinical radiobiologists and radiation therapists in view of the lack of standardized protocols for such clinical screenings, in order to adapt and personalize the therapeutic parameters to individual radiosensitivity of patients.

Some comments that should be addressed by the authors:

  1. Lines 48-49: The authors should provide an explanation and examples of how the radiotherapy treatment plan could be personalized and adapted to the individual radiation sensitivity to improve the therapeutic ratio. By adjusting the cumulative tumor dose, the dose per fraction, a change in dose rate....?
  2. Lines 119-121: The LET and thus the impact of different radiation qualities on the various biodosimetric endpoints should briefly be introduced and explained to the unprimed reader.
  3. Lines 122-130: The differences between transmissible aberrations and unstable, lethal aberrations should be explained in more detail. This also includes the relevance of the time point of analysis after exposure and the importance of the analysis of cytogenetic damage in 1st mitoses post-exposure. See also 10.
  4. Lines 131-133: Although the main focus is on cytogenetic markers, the impact of repair kinetics on the decay of DSB repair foci (‘strong limitations on a time scale’) should be explained and discussed briefly.
  5. Line 134: The authors may briefly explain the differences between the radiation response of lymphocytes and fibroblasts, the frequently described difference in aberration rates (impact of nuclear geometry?), and why lymphocytes are better suited for these predictive assays than fibroblasts apart from their availability.
  6. Line 174: The authors should include a short comment on the advantage of the PCC method and the differences between G1-PCCs (PEG fusion) and G2-PCCs (calyculin A).
  7. Line 228: The abbreviation ‘Tn’ for translocation has not been introduced (line 127?).
  8. Lines 244-245: Besides a compromised cell cycle regulation, increased radiosensitivity may be associated with impaired and imbalanced DSB repair pathways causing a shift from canonical non-homologous end-joining or homologous recombination (gene conversion) towards more error-prone mechanisms of DSB repair such as alternative-NHEJ/MMEJ or single-strand annealing. In my opinion, this is a relevant point that should be addressed by the authors.
  9. Line 246, Table 1: PCC spreads are G1-PCCs or G2-PCCs or both? This is only indicated by Borgmann et al. 2002 (‘fusion PCC’ = G1-PCCs).
  10. Table 1 (Borgmann et al. 2008) and Table legend: The importance of BrdU exposure and the discrimination between first and second mitoses post-exposure should be explained to the reader in more detail.
  11. Table 1 (Padjas et al. 2012): According to my knowledge, the common acronym for a medical linear accelerator is linac, not LinAc.
  12. Lines 278-279 plus line 493: The option of using caffeine or more specific Chk1 inhibitors to overcome the G2/M checkpoint for conventional metaphase analysis and the scoring of micronuclei should be mentioned and discussed, in particular concerning the ‘AHAP’-principle. The authors briefly mention this point in lines 499-501 when introducing the G2-PCC method.
  13. Section 3.4.1: Are there any indications that specific DSB-repair pathways and their impairment are related to NTT? In addition to gH2AX, other DSB surrogate markers (e.g. Rad51, BRCA1, etc.) could be mentioned that represent specific indicators for DSB repair pathways that may play a role in the development of NTT (e.g., https://doi.org/10.1371/journal.pone.0091319).
  14. General remark: The endpoints considered here all aim to unravel a relationship between impaired DNA repair and NTT. This may be due to a cell cycle-specific impairment of DSB repair pathways such as NHEJ or HR. Cell cycle-dependent analysis of DSB repair has proven to be increasingly important for elucidating mechanistic aspects of DSB repair and may also be crucial for the expression of clinical hypersensitivity to ionizing radiation, e.g. shown in https://doi.org/10.1016/j.dnarep.2020.102992.
  15. Another problem concerning cell cycle-specific and biphasic DSB repair kinetics pertains to the analysis of cytogenetic damage using the G2 assay, which is commonly conducted within the first hours after irradiation and therefore does not consider the efficacy of homologous recombination. Deficits in homologous recombination can therefore not be detected, since this repair mechanism only plays a role in the slow repair component in G2 6-8h after irradiation. This point should also be addressed and discussed in more detail by the authors.
